# Diagnosis and Management of Mucopolysaccharidosis Type II (Hunter Syndrome) in Poland

**DOI:** 10.3390/biomedicines11061668

**Published:** 2023-06-08

**Authors:** Zbigniew Żuber, Beata Kieć-Wilk, Łukasz Kałużny, Jolanta Wierzba, Anna Tylki-Szymańska

**Affiliations:** 1Department of Pediatrics, Faculty of Medicine and Health Sciences, Andrzej Frycz Modrzewski Krakow University, 30-705 Krakow, Poland; 2Unit of Rare Metabolic Diseases, Department of Metabolic Diseases, Jagiellonian University Medical College, 31-008 Krakow, Poland; 3Department of Pediatric Gastroenterology and Metabolic Diseases, Poznan University of Medical Sciences, 61-701 Poznan, Poland; 4Department of Pediatrics, Hematology and Oncology, Medical University of Gdansk, 80-210 Gdansk, Poland; 5Department of Pediatrics, Nutrition and Metabolic Diseases, The Children’s Memorial Health Institute, 04-730 Warsaw, Poland

**Keywords:** lysosomal storage disease, mucopolysaccharidosis II, Hunter syndrome, enzyme replacement therapy

## Abstract

Mucopolysaccharidosis type II (MPS II; also known as Hunter syndrome) is a rare, inherited lysosomal storage disease. The disease is caused by deficiency of the lysosomal enzyme iduronate-2-sulphatase (I2S) due to mutations in the *IDS* gene, which leads to accumulation of glycosaminoglycans (GAGs). Deficiency of I2S enzyme activity in patients with MPS II leads to progressive lysosomal storage of GAGs in the liver, spleen, heart, bones, joints, and respiratory tract. This process disturbs cellular functioning and leads to multisystemic disease manifestations. Symptoms and their time of onset differ among patients. Diagnosis of MPS II involves assessment of clinical features, biochemical parameters, and molecular characteristics. Life-long enzyme replacement therapy with idursulfase (recombinant human I2S) is the current standard of care. However, an interdisciplinary team of specialists is required to monitor and assess the patient’s condition to ensure optimal care. An increasing number of patients with this rare disease reach adulthood and old age. The transition from pediatric care to the adult healthcare system should be planned and carried out according to guidelines to ensure maximum benefit for the patient.

## 1. Introduction

Mucopolysaccharidosis type II (MPS II; also known as Hunter syndrome) is a rare X-linked, recessively inherited lysosomal storage disease. The disease is caused by deficiency of the lysosomal enzyme iduronate-2-sulfatase (I2S), which catalyzes the hydrolysis of 2-sulfate groups on dermatan sulfate and heparan sulfate.

This deficiency is caused by mutations in the *IDS* gene, which lead to lysosomal and extracellular accumulation of glycosaminoglycans (GAGs) [1,2,3]. MPS II is the most common mucopolysaccharidosis, with an estimated prevalence ranging from 0.13 to 2.16 per 100,000 live births in the European population [4]. In Poland, the estimated prevalence of MPS II is 0.46 per 100,000 live births [5]. As the disease is inherited in an X-linked recessive way, it almost always occurs in males. Carriers of the mutant *IDS* gene are asymptomatic but cases of affected female patients have been described [6,7].

Lower I2S enzyme activity in patients with MPS II leads to lysosomal storage of heparan sulfate and dermatan sulfate. Heparan sulfate tends to damage the central nervous system (CNS), whereas dermatan sulfate is responsible for connective tissue damage. This process leads to multisystemic disease manifestations [8]. Clinical diagnosis of MPS II is challenging, especially in the early stages of the disease, when symptoms are subtly expressed [9].

MPS II has two clinical forms: neuronopathic, with CNS involvement, and non-neuronopathic, without involvement of the CNS. In the non-neuronopathic form, accumulation of dermatan sulfate primarily occurs, sparing the CNS. The neuronopathic form is estimated to occur in two thirds of patients [10]. Patients usually have a higher birth weight than healthy newborns; somatic signs commonly appear between 18 and 24 months of age but the neuronopathic form may present earlier [11,12]. The non-neuronopathic (also called attenuated) form is characterized by slow progression of peripheral symptoms and absent or reduced cognitive problems [8]. Patients with non-neuronopathic MPS II usually survive until late adulthood [13].

The aim of this review is to describe actual methods of diagnosis, treatment, and management of patients with MPS II.

## 2. Diagnosis

Clinical diagnosis of MPS II relies on the recognition of subtle, nonspecific symptoms, especially in early childhood (up to 3 years of age) (Table 1) [3]. Diagnosis of MPS II involves assessment of clinical features, biochemical parameters, and molecular characteristics.

In MPS II, as an X-linked defect, we perform family screening only on request. Although nowadays the idea of neonatal screening is strongly emphasized, in many countries, it is still not in practical use [3].

Patients with slowly progressive (non-neuronopathic) disease show varying degrees of changes in different organs, which can make MPS II difficult to recognize [9]. Limited range of motion in the shoulder joints is an early symptom, and typical changes can be found on ultrasound of the hip joints. Coexistence of these symptoms with high birth weight and recurrent infections of the upper respiratory tract may indicate MPS II. The onset of symptoms usually occurs at preschool age. Earlier onset of clinical symptoms usually signifies a more severe clinical disease course [8,14].
biomedicines-11-01668-t001_Table 1Table 1Clinical symptoms of MPS II.FeatureSymptomsAnthropological abnormalitiesNormal or high birth weight and body length, growth inhibition from the age of 3 years and then short stature, growth deficiency, macrocephaly [12,15]Dysmorphic abnormalitiesThickened facial features, broad nose, large head circumference, thick hair with a low neckline [3,8]Rheumatological/orthopedic abnormalitiesReduced range of joint motion, *dysostosis multiplex* visible on X-ray examinations, carpal tunnel syndrome, ultrasonographic features of hip joints [16,17,18,19]Laryngological abnormalitiesEnlarged tongue, hypertrophic tonsils, slurred speech over time, gingival overgrowth, widely spaced teeth, recurrent middle-ear infections, progressive hearing loss [3,8]Pulmonological abnormalitiesRecurrent upper respiratory tract infections and pneumonia, progressive airway obstruction, sleep apnea [3,8]Cardiological abnormalitiesValvular thickening and regurgitation, cardiomyopathyGastrointestinal/surgical abnormalitiesLiver or liver and spleen enlargement, diarrhea, inguinal and umbilical hernias [3,8]Neurological and developmental abnormalitiesBehavioral disorders, poor/lack of concentration, hyperactivity, various degrees of mental retardation [3,8]


### Laboratory Tests

Clinical suspicion of MPS II should be confirmed using biochemical and genetic tests. 

Laboratory diagnosis of MPS II includes three steps [2,3]: Assessment of urinary GAG excretion;Analysis of I2S activity in plasma, leukocytes, and fibroblasts, with simultaneous determination of a second sulfatase;Identification of a pathological mutation in the *IDS* gene.

Methods using high-performance liquid chromatography (HPLC) with tandem mass spectrometry (HPLC-MS/MS) or electrophoresis have been developed to quickly and accurately quantify and qualify excreted GAGs in a single urine analysis. Liquid chromatography can distinguish between non-neuronopathic and neuronopathic MPS II subtypes. Determination of I2S activity can be performed using peripheral blood leukocytes, cultured skin fibroblasts, or dried blood spot samples. Since multiple sulfatase deficiency is also characterized by decreased I2S activity, it should be excluded by simultaneous measurement of other sulfatases to confirm I2S deficiency and the diagnosis of MPS II [20,21,22].

Diagnosis based on biochemical tests should be confirmed by identification of the pathogenic variant of *IDS*. Most mutations are point missense mutations, although many patients also have intragenic deletions or duplications and complex rearrangements that include the nearby *IDSP1* pseudogene. Therefore, assessment of mutation status in patients with suspected MPS II should include sequencing analysis as well as use of methods to detect deletions and rearrangements, such as multiplex ligation-dependent probe amplification or microarray [23,24].

## 3. Treatment

### 3.1. Enzyme Replacement Therapy

Enzyme replacement therapy with idursulfase (a recombinant human I2S that uses the mannose-6-phosphate receptor on the cell surface and lysosomes) is the current standard of care. Enzyme replacement therapy is administered via weekly intravenous infusion. Idursulfase does not affect the cognitive and behavioral functioning of patients with MPS II because it does not cross the blood–brain barrier [3,25,26]. 

The approval of idursulfase for long-term treatment of patients with MPS II was based on the results of a randomized, double-blind, placebo-controlled trial [27]. Idursulfase was administered weekly at a dose of 0.5 mg/kg body weight. After 1 year, idursulfase improved the range of joint motion and respiratory parameters. Urinary GAG levels were also significantly reduced from baseline in patients receiving enzyme replacement therapy compared with placebo. Idursulfase, administered weekly at a dose of 0.5 mg/kg, was approved for the treatment of MPS II in the European Union [27].

### 3.2. Treatment Guidelines

The method of treatment depends on the patient’s age at diagnosis and the stage and severity of disease:Children up to 2 years of age—hematopoietic stem cell transplantation (HSCT) or enzyme replacement therapy.

Early initiation of enzyme replacement therapy is of key importance as it offers a chance to slow down the disease process and can prevent the irreversible effects of the disease which progressively appear over time:Non-neuronopathic form without neurodevelopmental deficits or disorders—enzyme replacement therapy or conservative treatment, depending on the patient’s clinical status and/or the patient’s decision;Neuronopathic form (mildly advanced)—enzyme replacement therapy for a trial period, with follow-up for at least 1 year; however, based on current clinical experience, 2–3 years of follow-up are preferred;Neuronopathic form (moderately advanced)—enzyme replacement therapy for a short trial period, with follow-up for at least 1 year;Neuronopathic form (advanced)—conservative treatment (feeding with percutaneous endoscopic gastrostomy, no indications for ventilation).

### 3.3. Recommendations for the Initiation of Enzyme Replacement Therapy

The initiation of treatment should be preceded by a careful baseline assessment and setting of clear treatment goals to objectively assess the effect of treatment;Enzyme replacement therapy should be introduced as early as possible after diagnosis;All patients with neuronopathic MPS II without severe neurological symptoms (without current CNS involvement) should be treated with enzyme replacement therapy using idursulfase;Based on mutation analysis and/or family history of previously affected relatives, patients with neuronopathic disease may be eligible for enzyme replacement therapy on a case-by-case basis. Enzyme replacement therapy in this group can be considered only at an early stage of the disease before the onset of significant neurological disease. A clear discussion with the family regarding the ineffectiveness of enzyme replacement therapy in CNS disease is necessary and the family should be warned that enzyme replacement therapy will be terminated if significant CNS disease is detected in a child;The applied dose of idursulfase is fixed at 0.5 mg/kg body weight, administered intravenously every week;Treatment should be carried out under the supervision of specialists with experience and expertise in the treatment of MPS II and enzyme replacement therapy.

Before starting enzyme replacement therapy, discussions with the patient and/or their guardians is necessary. In the non-neuronopathic form of MPS II, there may be a subclinical period when symptoms are not yet strongly expressed. Opinions differ regarding the early implementation of enzyme replacement therapy, considering the need for weekly therapy, route of administration (intravenous), and duration of treatment (potentially for the rest of life). However, the disease is progressive and best treatment results were observed after early introduction of enzyme replacement therapy. Clinical trials showed that administration of enzyme replacement therapy within 12 months from starting therapy improved clinical parameters in patients with non-neuronopathic MPS II. Urinary excretion of GAGs decreased during the first months of treatment and remained reduced. Liver and spleen volumes were also reduced (by more than 30%), and this was sustained in all patients until the end of the study. Improvement was also noted in the 6 min walk test and percent predicted forced vital capacity, and the left ventricular mass index was reduced by more than 12%. The ejection fraction and the functioning status of the heart valves were stable. Range of motion increased from 8.1 to 19.0 degrees, depending on the joint [28].

Early treatment, especially before the development of advanced organ changes and secondary complications, can slow down the progression of the disease and improve the patient’s quality of life. If the patient or patient’s guardians wish to postpone, suspend, or not start enzyme replacement therapy, the doctor should monitor the patient’s condition as if the patient were being treated [3,29].

### 3.4. Limitations of Enzyme Replacement Therapy

Debates considering the management of patients with CNS involvement have been held worldwide and refer to criteria for the initiation and cessation of therapy. Since the timing of enzyme replacement therapy initiation is crucial, we suggest starting treatment as early as possible, without any age limits. During treatment, assessment every 6 months is required to ensure the consent of the Committee for the Treatment of Ultra Rare Diseases (CTURD) for treatment continuation. A comprehensive report on the course of treatment and results of follow-up is submitted by the treating physician. The CTURD is appointed by the President of the National Health Fund and decides to extend or withdraw consent for treatment continuation. If there is a significant neurological burden, the risks and potential benefits of therapy should be discussed among the family, physician, and payer [10,30].

### 3.5. Safety of Enzyme Replacement Therapy

The most common adverse events of idursulfase administration are infusion-related reactions, urticaria, fever, and headache. Most side effects are mild to moderate. Management of side effects includes slowing or interruption of the infusion, and use of antipyretics, antihistamines, and glucocorticosteroids. Serious anaphylactic reactions have been observed in some patients; enzyme replacement therapy is, therefore, administered under hospital conditions as part of one-day hospitalization, in accordance with the regulations in Poland [31].

### 3.6. Immunogenicity of Enzyme Replacement Therapy

About half (45–55%) of patients treated with enzyme replacement therapy have IgG antibodies to idursulfase. However, no association was found between the presence of these antibodies and the frequency of adverse events. If there are no clinical indications or worsening of the treatment effect, there is no need for routine determination and monitoring of anti-drug antibody levels in blood of patients receiving enzyme replacement therapy. Additionally, no specific IgE antibodies against idursulfase were detected, suggesting that immunoglobulins do not contribute to the activation of an allergic reaction [27]. 

In MPS II, as in other lysosomal storage diseases, recombinant enzymes administered intravenously often lead to the production of anti-drug antibodies. Neutralizing antibodies can impair the desired biological effects of the drug. Exogenous enzymes are recognized by the immune system as foreign proteins. These are taken up by antigen-presenting cells and then processed and presented to helper T cells, thereby activating the production of antibodies. The effects of the immune response can be complex. ADA can interfere with the biological activities of the therapeutic enzyme through various mechanisms: enzyme-altered targeting, increased enzyme turnover, inhibition of enzyme uptake by cells, and inhibition of its catalytic site. Therefore, measurement of neutralizing anti-drug antibodies should be considered in patients with MPS II, as they could lead to an increasingly weaker biological response to well-tolerated enzyme replacement therapies over time [32,33].

### 3.7. Other Therapies

Data on outcomes in patients with MPS II are limited because diagnosis is typically made later than in MPS I. In some cases, HSCT has decreased urinary GAG levels, normalized spleen and liver volume, and improved respiratory, visceral, and skeletal functions. Nevertheless, HSCT does not stop the progression of neurological symptoms [34,35]. 

Substrate reduction therapy is a promising method based on small-molecule inhibitors of GAG synthesis, which prevents substrate storage. Substrate replacement therapy is not specific to MPS II and may be applicable to all MPS types [36,37,38,39,40].

Only gene therapy could repair the underlying cause of enzyme deficiency. Studies in animal models showed reduced GAG levels in the brain and improved neurological outcomes. Currently, clinical trials of gene therapy for MPS are ongoing worldwide [41,42].

### 3.8. Managing a Patient with MPS II during Enzyme Replacement Therapy

Idursulfase is indicated for the long-term treatment of patients with MPS II, both during maturation and after reaching adulthood [3]. A patient receiving enzyme replacement therapy (ERT) should remain under the care of the reference center. They should receive comprehensive monitoring with appropriate examinations determined by the Drug Program in Poland [43].

The results of randomized clinical trials showed the improvement of the range of joint motion in patients with MPS II (patients undergoing therapy retain the ability to actively move for a longer time and obtain an improvement in the range of abduction and flexion movements, specifically for the shoulder joint). Furthermore, ERT therapy leads to stabilization of respiratory parameters and reduction in the relative size of the liver and spleen. The subjective assessment of the quality of life of patients, in the opinion of themselves and their parents/guardians, either improves or stabilizes during the therapy. Moreover, urinary GAG levels were significantly lower than baseline in patients receiving treatment compared to placebo [27]. The observations clearly emphasize the importance of the timing of ERT initiation, even in childhood and in milder forms of MPS II. ERT effectiveness started in adulthood does not bring such clear-cut benefits [28,29,30].

According to the guidelines, ERT is life-long therapy. However, it requires the interdisciplinary team of specialists that will be monitoring and assessing the patient’s condition to ensure optimal patient care.

## 4. Co-Ordinated Care for a Patient with MPS II

Due to the complex nature of the disease, several organs need to be monitored and kept under control by the pediatrician or internist. Supplement S1 presents a set of examinations enabling comprehensive monitoring of patients with MPS II in Poland [43] Table 2.

## 5. Transition to Adult Health Care

A growing number of pediatric patients with rare, previously fatal diseases nowadays can reach adulthood, which is generally considered as a therapeutic success. However, despite this undeniable proof of a progress in rare disease patient care, several studies have shown that, without adequate support in the period of adolescence and early adulthood, the transition from the pediatric to the adult healthcare system is associated with an increased risk of deterioration in patients’ health and quality of life, assessed in the health-related quality of life (HRQOL) questionnaire [44,45].

The transition step should be a structured process targeted into creating a system of continuity and co-ordinated medical care adapted to the adolescent’s mental and social development. In addition, it should encourage the empowerment and self-determination of the patient within his family, society, and professional life [46]. In Poland, there are no official guidelines for the organization of this process, neither for the civilization diseases and other chronic diseases, nor for the rare diseases. The patient reaching 18 year of age is transferred directly to the adult care unit, without any preparation, plan, or adaptation visits, whereas, in other countries, a dedicated procedure already exists [47,48,49], including European Parliament Directive (2014) highlighting the patient transition organization as an important area and particular responsibility of European Reference Networks (ERN) centers [50].

## 6. Organization of the Patient Transfer Process

The transition process from pediatric care to the healthcare unit for adults should consider three main factors: location of the patient in the healthcare system, treating centers’ experience in the particular disease management and the correct education of the patient and his family, and being confronted with patients’ developmental changes typical for an adolescence period.

Medical personnel who have been taking care of a patient during his childhood should be trained to provide the adolescent patient and his caregivers with accessible and clear information as such a transition step is necessary and inevitable in the natural course of patient’s journey. It is important to control medical information flow between medical staff so that the physician who is taking over the care of the patient is equipped with a complete knowledge about the patient’s history. Therefore, accurate preparation of medical documentation containing key information on the patient’s current health condition, therapeutic recommendations, and algorithms for management in the case of a life-threatening event is a crucial step to ensure proper information flow.

It seems necessary to arrange mutual visit(s) to enable both the adult care and pediatric care physicians to share the disease history and the specificity of the patient’s condition. This will ensure that all of the medical staff engaged in the process are aware of the crucial therapeutic aspects, including the possible event of patient’s condition deterioration. Additionally, a separate ‘take-over’ visit including both treating physicians and the patient (and his family, if needed) can minimize stress connected with healthcare provider change. 

## 7. Patient Associations 

The activity of the Polish Association of patients with MPS and rare diseases [51] cannot be overestimated. The Association not only enables support and integration of patients and their families, but also acts as a support group at various stages of disease development. In Poland, the Association has been actively working since 1990, organizing rehabilitation camps each year as well as international conferences with the participation of specialists, which enables patients to obtain consultations and provides training for new medical staff. 

## 8. Conclusions

Due to a recent advance in MPS2 awareness among healthcare professionals in Poland, resulting in a decreased time of the accurate diagnosis confirmation, patients can experience benefits of early treatment initiation and complex, multidisciplinary care run in specialized centers. Therefore, an increasing number of patients suffering from this condition may reach adulthood, which implicates the necessity of their coordinated transition to the adults’ health care units, specialized in rare disease management. Along with a facilitated access to additional services, i.e., rehabilitation and psychological and family support, a standardized transition process still requires effective implementation in Poland. Another limitation in early access to enzyme replacement therapy is still the patient’s age required to qualify to National Drug Program (5 y. o. despite no age limit according to SmPC). 

The program of treatment and care presented in this article is a subject of constant analysis and adaptation to evolving circumstances, in order to improve the health conditions of Polish MPS2 patients.

## Figures and Tables

**Table 2 biomedicines-11-01668-t002:** Comprehensive monitoring of the patients with MPS II in Poland.

Tests Performed during the Qualification
Blood count Coagulation screen Proteinogram Gasometry AspAT, AlAT, CK, bilirubin Lipidogram Vitamin D and K Urinary excretion of mucopolysaccharides The level of antibodies to idursulfase (not obligatory) Vital signs Anthropometric measurements EEG EKG Echocardiography Chest X-ray Spine X-ray Abdominal ultrasound MRI of the CNS, including the cervical spine Diagnosis of carpal tunnel (EMG) Pulmonary examination (respiratory efficiency, spirometry) Audiometric examination Orthopedic examination with assessment of joint mobility Ophthalmological examination Psychological examination with the IQ assessment or psychomotor development in younger children Examination of the musculoskeletal system and motor functions 3/6 min walk test SF36 test
Treatment monitoring (during the first year of treatment, examinations performed every 6 months)
During the first year of treatment, examinations should be performed every 6 months Blood count Coagulation screen Gasometry AspAT, AlAT, CK, bilirubin Urinary excretion of mucopolysaccharides The level of antibodies to idursulfase (not obligatory) Vital signs Anthropometric measurements EEG EKG Echocardiography Abdominal ultrasound Pulmonary examination (respiratory efficiency, spirometry) Orthopedic examination with assessment of joint mobility Examination of the musculoskeletal system and motor functions 3/6 min walk test SF36 test An impartial physician not involved in the treatment of patients with Hunter disease should periodically evaluate the effectiveness of therapy. The treatment can be extended every 6 months upon the decision of the Co-ordination Team for Ultra Rare Diseases, based on the Therapy Monitoring Protocol provided by the treating physician.
Treatment monitoring (examinations performed every 365 days)
Lipidogram Vitamin D and K Chest X-ray Spine X-ray MRI of CNS (primarily indicated in the case of concomitant hydrocephalus, depending on the doctor’s decision) Audiometric examination Ophthalmological examination with the fundus assessment Diagnosis of carpal tunnel (EMG)
Program monitoring
Collecting data on treatment monitoring in the patient’s medical records and presenting them each time at the request of the controllers of the National Health Fund Supplementing the data contained in the register (SMPT) available via the Internet application provided by the National Health Fund, with the frequency consistent with the description of the program and at the end of the treatment Submitting reporting and billing information to the National Health Fund: information is provided to the National Health Fund in paper or electronic form, in accordance with the requirements published by the National Health Fund.

## Data Availability

Not available.

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
