# Peer review of "Diagnosis and Management of Mucopolysaccharidosis Type II (Hunter Syndrome) in Poland"

_biomedicines, 2023, doi:10.3390/biomedicines11061668_

Round 1

Reviewer 1 Report

please, see attachment

Author Response

QUESTION: The review article entitled “Diagnosis and management of mucopolysaccharidosis type II 2 (Hunter syndrome) in Poland” by Zbigniew Żuber and co-authors describes the current methods of diagnosis for the management and treatment of patients affected by MPS II.

The article has a good structure and a proper logical flow. English language is appropriate.

In the title the focus of your article is on the diagnosis and management of MPSII in Poland. However, it seems a generic description of techniques and tests for MPSII. Probably, concepts and information more focused on Poland should be reiterated in the manuscript.

The manuscript misses of figures. Adding at least one figure would improve the value of your article. I suggest to prepare one figure with the chemical structures of heparan and dermatan sulphates, or with the MS/MS spectra or with the chromatographic profiles of the different GAGs.

ANSWER: Algorithms for the diagnosis and therapy of patients with MPS2 are included in the articles references [3,23]. They are very well developed, so we do not see the need to construct another scheme that would basically coincide with those previously published

Q: The introduction should contain more details about the I2S protein. This may include expression levels, structure, activities etc.

A: This was included in the Introduction lines 34-36.

Q: The introduction should better explain the brain features and the neurological impairment in MPS II. Furthermore, as comparison, the background misses of important references related to the neurologic and metabolic disturbances in other MPS (such as MPSIII), in which these alterations were explained by alterations in proteins involved in cytoskeleton organization, vesicle trafficking and fatty acid metabolism (please use the following refs: 10.3390/biom10030355; 10.3390/ijms21124211).

A: Thank you for the interesting suggestions, however, the message of our work concerns not so much theoretical issues as the practical aspect of diagnosing and treating patients with MPS II.

Q: The Diagnosis section (or the introduction for a more general description) misses of an important recall to the expanded newborn screening programs worldwide, which include laboratory analyses to detect the biochemical/molecular defects of some MPS types in order to perform early diagnosis at the neonatal age.

A: It was added as advised:

In MPS II, as an x-linked defect, we perform family screening only on request. Although nowadays the idea of neonatal screening is strongly emphasized, in many countries it is still not in practical use [3].

Q: Line 81: “ultra-efficient liquid chromatography” does not exist. Please, replace with high-performance liquid chromatography (HPLC).

A: corrected as advised

Q: Line 83: replace “quantify and qualify” with “identify and quantify”

A: corrected as advised

Q: Line 83-84: the authors comment that “Liquid chromatography can distinguish between non-neuronopathic and neuronopathic MPS II sub-types.”. What do they exactly refer to? To the ability of HPLC to discriminate retention times of different GAGs? This should be better explained.

A: it was quoted in Zhang, H.; Wood, T.; Young, S.P.; Millington, D.S. A straightforward, quantitative ultra-performance liquid chromatography-tandem mass spectrometric method for heparan sulfate, dermatan sulfate and chondroitin sulfate in urine: an improved clinical screening test for the mucopolysaccharidoses. Mol. Genet. Metab. 2015, 114, 123–128. [20]

Q: Line 89-90: Please, rephrase the following sentence for a better readability: “Since multiple sulfatase deficiency is also characterized by decreased I2S activity, it should be excluded by simultaneous measurement of other sulfatases to confirm I2S deficiency and the diagnosis of MPS II”

A: corrected as advised

Q: Line 104-105: “Idursulfase does not affect the cognitive and behavioral functioning of patients with MPS II because it does not cross the blood-brain barrier” means that the treatment does not ameliorate the cognitive functions? Indeed, we mean it,

A: ERT has a soothing effect on somatic symptoms only.

Deficiency of the activity of idursulphatase leads to the accumulation of dermatan sulphate and heparan sulphate. Heparan sulphate has a damaging effect on the central nervous system.

Q: Line 113: The “Treatment guidelines” continues with a series of bullet points that do not explain if there are specific guidelines depending on the age and severity of patients (as it is suggested by the authors in the first sentence at line 114). This should be better clarified.

A: This is discussed in the paragraph 3.3. Recommendations for the initiation of enzyme replacement therapy.

Q: Line 128: “Recommendations for the initiation of enzyme replacement therapy”. An introductive one-line sentence to explain the value or importance for the existence of these recommendations should be added to help the reader to understand the story.

A: Sentence was added:

Early initiation of treatment is of key enzymatic importance as it offers a chance to slow down the disease process and can prevent the irreversible effects of the disease which progressively appear over time.

Q: Line 231: ERT abbreviation was not added next to the enzyme replacement therapy when it was firstly cited in the text.

A: It was added as advised

Reviewer 2 Report

This review paper titled “Diagnosis and management of mucopolysaccharidosis type II (Hunter syndrome) in Poland” by Tylki-Szymańska et al highlights the significance of an interdisciplinary team in monitoring and assessing the patient’s condition for optimal care. However, there are some areas where this paper needs improvement.

1.                   In the Introduction; Line 44-27 needs a reference demonstrating HS affects CNS  and DS target the connective tissues, also give  examples  of such tissues,i.e.,  lung, liver, spleen, lymph node bone, etc.

2.                   In the section on diagnosis, it may be helpful to include a flowchart or algorithm to aid clinicians in the diagnostic process.

3.                    While Table 1 lists the clinical manifestations of Hunter syndrome, it should include the symptoms related to liver and spleen defects and matching references for each row in a new third column. The title of Table 1 should not include the references.

4.                   Furthermore, the conclusion should include a paragraph that describes the limitations and side effects of current therapies, such as infusion-related immune and non-immune reactions, as indicated in earlier sections.  Additionally, it  is suggested to include future directions for research and potential areas for improvement in the management of Hunter syndrome.

Author Response

QUESTION: This review paper titled “Diagnosis and management of mucopolysaccharidosis type II (Hunter syndrome) in Poland” by Tylki-Szymańska et al highlights the significance of an interdisciplinary team in monitoring and assessing the patient’s condition for optimal care. However, there are some areas where this paper needs improvement.

  1. In the Introduction; Line 44-27 needs a reference demonstrating HS affects CNS  and DS target the connective tissues, also give  examples  of such tissues,i.e.,  lung, liver, spleen, lymph node bone, etc.

ANSWER: It was quoted in Mohamed, S.; He, Q.Q.; Singh, A.A.; Ferro, V. Mucopolysaccharidosis type II (Hunter syndrome): clinical and biochemical aspects of the disease and approaches to its diagnosis and treatment. Adv. Carbohydr. Chem. Biochem. 2020, 77, 71–117 [2]

  1. In the section on diagnosis, it may be helpful to include a flowchart or algorithm to aid clinicians in the diagnostic process.

ANSWER: Algorithms for the diagnosis and therapy of patients with MPS2 are included in the articles references [3,23]. They are very well developed, so we do not see the need to construct another scheme that would basically coincide with those previously published

Q: While Table 1 lists the clinical manifestations of Hunter syndrome, it should include the symptoms related to liver and spleen defects and matching references for each row in a new third column.

A: added as advised

The title of Table 1 should not include the references.

A: References from the title of table 1 have been removed. Corrected as advised

Q: Furthermore, the conclusion should include a paragraph that describes the limitations and side effects of current therapies, such as infusion-related immune and non-immune reactions, as indicated in earlier sections.  Additionally, it  is suggested to include future directions for research and potential areas for improvement in the management of Hunter syndrome.

A: These issues are covered in separate paragraphs: 3.4, 3.5, 3.6, 3.7.

Reviewer 3 Report

The authors would like to review mucopolysaccharidosis type II (MPS II; also known as Hunter syndrome) in Poland, but in the aim, they state to review in general. What could make the difference between this paper with the literature data I came across is the specific situation in the different provinces of Poland. The authors should add the ophthalmoscopic evaluation, which is missing, and provide figures of progress in Poland with additional figures of statistics of MPS II in the 90s until the end of the second decade of the 21st century. Otherwise, there is not much interest in reading your paper. We would like to have your Polish situation emphasized and collect an elevated number of citations. 

Author Response

Q: The authors would like to review mucopolysaccharidosis type II (MPS II; also known as Hunter syndrome) in Poland, but in the aim, they state to review in general. What could make the difference between this paper with the literature data I came across is the specific situation in the different provinces of Poland.

A: The aim of the study was to present and share Polish experience in diagnosing and treating patients with Hunter syndrome. The algorithm for managing and caring for patients with MPSII may be similar to that of other centers, but it is not identical, so we wanted to share it to enrich the experience pool for this condition.

Q: The authors should add the ophthalmoscopic evaluation, which is missing,

A: Unfortunately, we do not have the results of ophthalmological examinations in our patients.

Q (..) and provide figures of progress in Poland with additional figures of statistics of MPS II in the 90s until the end of the second decade of the 21st century. Otherwise, there is not much interest in reading your paper. We would like to have your Polish situation emphasized and collect an elevated number of citations. 

A: In Poland we started enzyme treatment in 2009, before that patients were monitored in one national center at the Children's Memorial Health Institute in Warsaw. We are unable to add more information and data that might be of interest to readers.

Round 2

Reviewer 3 Report

The authors properly addressed the comments and suggestions of the reviewers.